

# The Role of the Radial Vorticity Gradient in Intensification of Tropical Cyclones

Samuel Watson[1] and Courtney Quinn[1]

[1]School of Natural Sciences, University of Tasmania, Churchill Avenue, Sandy Bay, 7001, Tasmania, Australia

**Correspondence:** Courtney Quinn (courtney.quinn@utas.edu.au)

**Abstract.** The role of the radial vorticity gradient in tropical cyclone dynamics is explored through a low–order conceptual box model. Specifically, we look at stable–to–stable state transitions which may be linked to tropical cyclone intensification, dissipation, or eyewall replacement cycles. To this end, we identify two parameters of interest: the exponent of radial decline and sea surface temperature. We examine how variation in these parameters affect the stable states of the model and consider

the behaviour of the system under time–dependent parameters. By externally forcing the exponent of radial decline and sea surface temperature we show the existence of rate–dependent behaviour in the model. These findings are brought together in a case study of Hurricane Irma (2017). The results highlight the role of the radial vorticity gradient in behaviour such as rate–induced tipping and overshoot recovery. They also show that a simple model can be used to explore relatively complex tropical cyclone dynamics.

## 10   1   Introduction

Rapidly rotating storm systems, commonly called tropical cyclones (TCs), are one of the most iconic yet destructive atmospheric phenomena. It is estimated that the damage from TCs has resulted in an average cost of USD 51.5 billion over the last decade (Krichene et al., 2023). Improvements in our ability to accurately model and predict their behaviour will result in the saving of lives and infrastructure. In the last 50 years, much work has been done to develop systems of equations which

describe the fluid mechanical and thermodynamic evolution of TC systems, e.g. (Anthes, 1982; Emanuel, 1988). More recently, such systems have been adapted for analysis within a dynamical systems framework which has helped to extended our understanding of the qualitative nature of TCs, e.g. (Schönemann and Frisius, 2012; Slyman et al., 2023). Here, we use a dynamical system directly derived from first principles of physics to explore the role of the radial vorticity gradient in TC intensification, dissipation, and eyewall replacement cycles.

## 20   1.1   Vorticity, Intensification, and Eyewall Replacement Cycles

The circulation of a TC is initiated by combined vorticity effects in the atmosphere. This vorticity is composed of both the ambient vorticity due to the Earth's rotation, given by the Coriolis parameter, and the relative vorticity of the atmospheric flow. Weak vertical wind shear is also necessary so that a developing TC does not break apart as its convection grows through the layers of the atmosphere (Gray, 1998). Once initiated, TCs are maintained by convection within the eyewall, thus they require



a constant heat source. This heat is mainly provided through heat exchange between the ocean surface and the boundary layer flow. The observationally derived critical SST temperature for TC formation is $26.5^oC$; below this threshold TCs are not observed to form (Anthes, 1982).

An important phenomena which can occur within TCs is the eyewall replacement cycle (ERC), where a secondary tangential wind maximum forms outside the primary eyewall. As this secondary eyewall forms it is theorised that its convection consumes

an increasing proportion of the radial inflow, effectively 'choking' the inner eyewall (Kepert, 2013). As the inner eyewall dissipates the secondary eyewall contracts and intensifies. The sequence generally occurs over a 12 to 36 hour period and can repeat multiple times throughout the lifespan of the TC (Sitkowski et al., 2011). An ERC can impact TC forecasting as they cause the intensity of the TC to fluctuate dramatically. If the wind speed of the inner eyewall is used to measure the strength of a TC, an ERC may be mistaken for the dissipation of the TC.

ERCs were first observed occurring within Typhoon Sarah, which moved through the northern Pacific in 1956 (Fortner Jr, 1958). As observation techniques progressed, ERCs were found to be a common feature of TCs (Sitkowski et al., 2011). A recent example is the ERCs observed within Hurricane Irma, which formed in the North Atlantic in 2017 (Fischer et al., 2020). Two ERCs were observed with each taking place over less than 12 hours. Interestingly, the first ERC consisted of inner eyewall weakening and dissipation as expected, but the second ERC resulted in a continual rapid intensification of the TC (Fischer

et al., 2020). This behaviour contradicts the common model of ERCs and shows that there is still much work to be done in understanding this phenomenon.

By converging air, and thus energy, to the TC centre, the boundary layer plays an important role in intensity changes such as intensification or secondary eyewall formation. Frictional updraft plays a major role in driving convection within the eyewall and begins within the boundary layer (Kepert, 2013). Thus, changes to boundary layer parameters linked to frictional

updraft may lead to the strengthening or initiation of deep convection. Boundary layer parameters which have been proposed to directly affect convection within the eyewall are the inflow rate, tangential and gradient wind maxima, and the radial vorticity gradient (Kepert, 2013). Further, it has been theorised that the frictionally forced vertical velocity at an eyewall is aproximately proportional to the radial vorticity gradient (Kepert, 2013). In light of the model used in this study, we focus here on the closely related roles of the radial vorticity gradient and the inflow rate. If we consider an increase in the radial vorticity gradient, this

implies an increase in the radial wind shear, i.e. a greater tangential wind gradient. An increase in the tangential wind gradient can be explained by increased tangential wind speed within the eyewall, which implies the total energy of the TC has increased. An energy increase necessitates that energy transportation via the boundary layer flow increases, which can be realised by an increase in the radial inflow (Anthes, 1982). The relation can be framed in the opposite direction as it is not known that one change necessarily precedes the other.

Multiple studies have supported this theory regarding the connection between the radial vorticity gradient and TC intensification. In a study of TC development, Ge et al. (2015) found that the radial profile of the inner–core relative vorticity was important in determining the strength and success of initial TC intensification. They compared two vortex models with different inner–core structures and found that the vortex with a "higher inner–core vorticity and larger negative radial vorticity gradient" promoted the formation of small–scale convective cells which act to intensify the TC. Additionally, in an analysis of three TC





boundary layer models, Kepert (2013) found that a "relatively weak local enhancement" of the radial vorticity gradient outside of the eyewall can produce a frictional updraft of the strength necessary for initiation of a secondary eyewall. He also found that once a secondary eyewall formed it possessed significantly stronger frictional updraft than the inner eyewall due to its position within a lower vorticity environment.

## 1.2 Rate–Induced Dynamics

There are multiple mechanisms which can cause a system to transition between stable states. The most widely known and studied is *bifurcation–induced* transitions (or b–tipping). These occur when external forcing causes a system to cross a bifurcation (a point of change in local stability). Once the bifurcation point has been crossed, the stable state which the system had previously been tracking disappears, thus it must transition to a new stable state (assuming one exists).

Another, more recently discovered, mechanism for transitions is *rate–induced* transitions (or r–tipping) (Ashwin et al., 2012).
These transitions occur without the crossing of a bifurcation point and are instead a result of the rate at which a parameter is externally forced. When the rate of change of the external forcing profile is greater than some critical value, the system becomes unable to track its original stable state. Tipping occurs when the system moves too far from the stable state and crosses some threshold. These rate–dependent behaviours have been shown to exist in conceptual models of geophysical systems such as the Indian summer monsoon (Ritchie et al., 2019) and more recently TC formation (Slyman et al., 2023). As TCs experience
changing environmental conditions throughout their lifetime, rate–dependent behaviours can be expected to play a role in TC intensification as well.

To define the rate of external forcing we follow Ritchie et al. (2023) and define an external forcing parameter $\sigma \equiv \sigma(\lambda t)$, where $u = \lambda t$ is dimensionless. The rate parameter $\lambda$ has units per second/day/year/ect. depending on the application. It is useful to note that the rate of change of the external forcing parameter and the rate parameter are related by

$$\frac{\mathrm{d}\sigma}{\mathrm{d}t} = \frac{\mathrm{d}\sigma}{\mathrm{d}u}\frac{\mathrm{d}u}{\mathrm{d}t} = \lambda\frac{\mathrm{d}\sigma(u)}{\mathrm{d}u}, \tag{1}$$

which has units of $\sigma$ per second/day/year/ect. and depends on $\lambda$ and $\sigma(u)$ itself (Ritchie et al., 2023). So, for a fixed forcing profile $\sigma(u)$, $\lambda$ quantifies the rate of change of this profile. The critical rate is then the value of $\lambda$ at which rate–induced tipping occurs, assuming the magnitude of the parameter shift remains the same.

A phenomenon closely related to rate–induced tipping is the overshooting of a bifurcation point without tipping, sometimes
referred to as *return tipping* (Ritchie et al., 2023). In this case, the system is externally forced across a bifurcation but may avoid tipping and recover its original state if the reversal in the forcing is faster than some critical rate.

## 2 A Low–Order Model

This analysis uses a low–order box model derived from geophysical principles as presented by Schönemann and Frisius (2012) (S&F). The model uses cylindrical coordinates and assumes an axisymmetric TC, thus only considers variation in the radial ($r$)
and vertical ($z$) directions. It assumes a length scale over which variation of the Coriolis parameter is negligible and thus takes



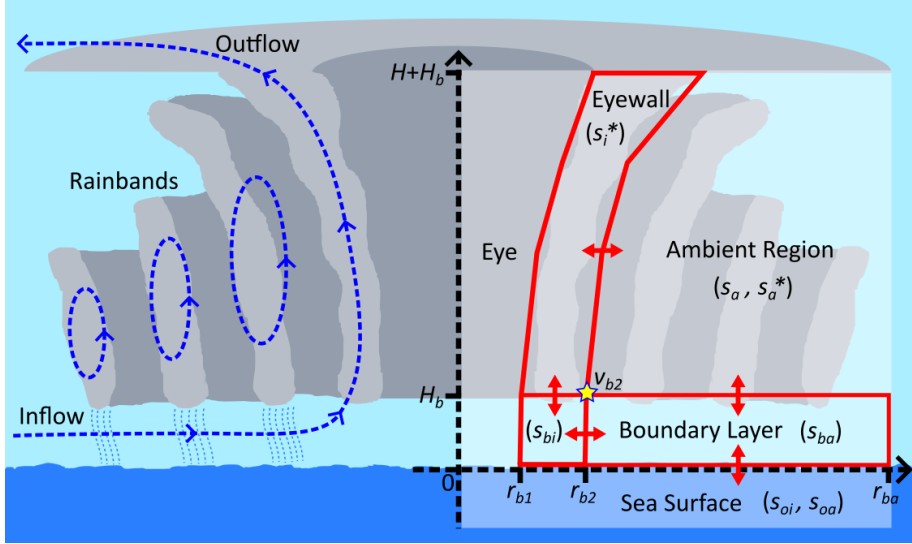

**Figure 1.** An idealised cross section of a TC with the box model overlaid (red). Here $H_b$ is the boundary layer height, $H$ the tropopause height, $s$ the specific entropy, and $r$ the physical radius in the boundary layer. The subscripts $i$, $b$, and $a$ denote the inner, boundary layer, and ambient region, respectively. The symbol $*$ denotes a variable at saturation.

it to be constant (f–plane approximation). It also applies both the Boussinesq and hydrostatic approximations to the governing fluid equations. The model considers three regions situated on top of a boundary layer; the eye, eyewall, and ambient region. A sketch of this division is shown in Figure 1. The boundaries between these regions are defined by lines of constant potential radius. The potential radius is the physical radius to which a particle must be moved, whilst conserving absolute angular momentum, in order bring its relative angular momentum to zero. The S&F model takes the potential radius to be defined as

$$R = \sqrt{r^2 + \frac{2vr}{f}} = \sqrt{\frac{2m}{f}}. \tag{2}$$

Here $v$ is the tangential wind velocity, $f$ the Coriolis parameter, and $m$ the angular momentum per unit mass. Hence, lines of constant potential radius correspond to lines of constant angular momentum, or 'angular momentum surfaces' in three dimensions.

The S&F model consists of a system of three first order non–linear differential equations which model the change in entropy within the important TC regions. In attempting to simulate the model we found inconsistencies in the time scales as written in the original paper. After conducting a thorough scale analysis we amended the model as follows:

$$\frac{ds_i^*}{dt} = A \left( \Psi_{b2}(s_i^*) \frac{s_{bi} - s_i^*}{M_i} \right) + \frac{s_a^* - s_i^*}{\tau_E}, \tag{3a}$$

$$\frac{ds_{bi}}{dt} = A \left( \Psi_{b2}(s_i^*) \frac{s_{ba} - s_{bi}}{M_{bi}(s_i^*)} + \frac{C_H}{2H_b} \left( |v_{b1}(s_i^*)| + |v_{b2}(s_i^*)| \right) \left( s_{oi}(s_i^*) - s_{bi} \right) \right), \tag{3b}$$

$$\frac{ds_{ba}}{dt} = A \left( \Psi_{b2}(s_i^*) \frac{\delta s_a - s_{ba}}{M_{ba}(s_i^*)} + \frac{C_H}{2H_b} |v_{b2}(s_i^*)| \left( s_{oa}(s_i^*) - s_{ba} \right) \right) + \frac{s_a - s_{ba}}{\tau_C}, \tag{3c}$$



where $A$ is the added rescale factor and set as $A = 3600$. Here $*$ denotes a variable at saturation. We have the dependent entropy variables corresponding to different regions of the model: $s_i^*$ is the eyewall saturated specific entropy, $s_{bi}$ is the eyewall boundary layer specific entropy, and $s_{ba}$ is the ambient region boundary layer specific entropy. The remaining entropy variables are as follows: $s_a^*$ is the ambient region saturated specific entropy, $s_{oi}$ is the sea surface specific entropy under the eyewall, and $s_{oa}$ is the sea surface specific entropy under the ambient region. These are either constants or functions of the dependent variables. It is important to note that these entropy variables are measuring the perturbation from the mean atmospheric entropy and are not a total entropy measure. From here on, 'entropy' refers to specific entropy. The mass stream function, $\Psi_2$, is responsible for mass exchange between regions. The region masses, $M$, with corresponding subscripts, denote the mass enclosed by each region. We then have constants $H_b$ – the height of the boundary layer, $\tau_E$ – the timescale of diabatic cooling, $\tau_C$ – the timescale of convective exchange, $\delta$ – the entrainment parameter (a proxy for the effects of wind shear), and $C_H$ – the transfer coefficient for enthalpy. The tangential velocities $v$ are taken at the inner ($b1$) and outer ($b2$) edges of the eyewall boundary layer. An outline of the auxiliary equations is provided in Appendix A1.

As a brief overview, for the change in eyewall entropy (3a) the first term on the RHS gives the vertical transport of entropy from the eyewall boundary layer into the eyewall. The second term gives the change due to diabatic cooling (heat exchange) between the eyewall and ambient region. For the change in eyewall boundary layer entropy (3b) the first term give the advective transport of entropy within the boundary layer, i.e. the horizontal transport from the ambient region boundary layer into the eyewall boundary layer. The second term gives the surface transfer of latent heat from the sea surface into the eyewall boundary layer. For the change in ambient region boundary layer entropy (3c) the first term gives the vertical transport of entropy from the ambient region into the ambient region boundary layer. The second term gives the surface transfer of latent heat from the sea surface into the ambient region boundary layer. The third term gives the entropy exchange due to shallow convection between the ambient and ambient boundary layer regions.

In this study we scale (3) to evolve on a timescale similar to that observed for the phenomena we are interested in. In the case of ERCs we are interested in timescales of 10-20 hrs. Via experimentation we scale (3) by a factor of 40. The constant parameter values used in this study are those given by S&F (2012) and are provided in Appendix A2.

In a physical context we are interested in the maximum wind speed a TC produces, thus here we present only the resulting tangential wind speed taken at the outer eyewall boundary, denoted $v_{b2}$. This wind speed is given as

$$v_{b2} = \frac{f}{2}\left(\frac{R_2^2 - r_{b2}^2}{r_{b2}}\right), \tag{4}$$

where $R_2$ and $r_{b2}$ are taken at the outer eyewall boundary.

To quantify changes to the radial vorticity gradient we first consider the absolute vorticity in the boundary layer, $\zeta_b$, which is composed of the ambient vorticity due to the rotation of the earth ($f$) and the relative vorticity of the fluid flow itself, given as

$$\zeta_b = f + \frac{v_b}{r_b} + \frac{\partial v_b}{\partial r_b}. \tag{5}$$





To determine the absolute vorticity at the outer eyewall boundary ($\zeta_{b2}$) a tangential wind profile (in the radial direction) for that region is needed. The model assumes a profile of

$$v_b = \frac{v_{b2} r_{b2}^{\beta}}{r_b^{\beta}} \quad \text{for} \quad r_b > r_{b2}, \tag{6}$$

where $\beta$ is called the *exponent of radial decline* and is given a physically relevant value between $0.5$ and $1$. Combining equations (5) and (6) and evaluating at $b = b2$ gives

$$\zeta_{b2} = f + (1 - \beta)\frac{v_{b2}}{r_{b2}}. \tag{7}$$

Thus, the radial vorticity gradient (evaluated at the outer eyewall boundary) is

$$\frac{\partial \zeta_{b2}}{\partial r_{b2}} = (\beta - 1)\frac{v_{b2}}{r_{b2}^2}. \tag{8}$$

While the physically plausible range for $\beta$ is $0.5 < \beta < 1$, it was found that for $\beta \approx 0.5$ the radial inflow was too weak to produce realistic maximum tangential winds and for $\beta = 1$ the radial inflow was unrealistically high (Schönemann and Frisius, 2012). The parameter $\beta$ illustrates the connection between the radial inflow and radial vorticity gradient as discussed in Sect.1.1. The focus of this study is to show the effect of variation in $\beta$ on TC dynamics. The variation of SST will also be considered due to the natural assumption that the underlying heat source will change as the TC propagates across the ocean.

## 3 Results

### 3.1 Stable States of the Model

Here we examine the physically plausible equlibria (stationary states) of the model. An equilibrium of the model represents a state where the rate of change of the entropy of each box is zero. Thus, equilibria can be identified as states of constant tangential wind. For our chosen parameter values, the model is characterised by four physically plausible equilibria: a 'rest
state', 'low wind', 'mid wind', and 'high wind'. The rest state is unstable (due to underlying assumptions in S&F (2012)) and corresponds to the system with no circulation. The low wind state is stable and corresponds to a low intensity circulating system. Such a system would be considered too weak to constitute a TC and is instead best interpreted as a tropical depression. The mid wind state is unstable and corresponds to a system moving from the tropical depression to TC state. The high wind state is stable and corresponds to a strong TC system.

These stable states change as the model parameters vary. We consider variation in $\beta$ and SST, and the changing equilibria can be tracked to produce a bifurcation diagram as shown in Figure 2. The equilibria mostly lose local stability through saddle-node bifurcation points. It should be noted that there is also a small region for low SST where the low wind state becomes unstable via a Hopf bifurcation – this corresponds to a region where no stable circulatory system is possible.

The region of $\beta$–SST parameter space of interest for intensification or ERCs is the bistable region where both the low and
high wind states exist. In this region and at its boundary, there is the possibility of transitioning between these states. These



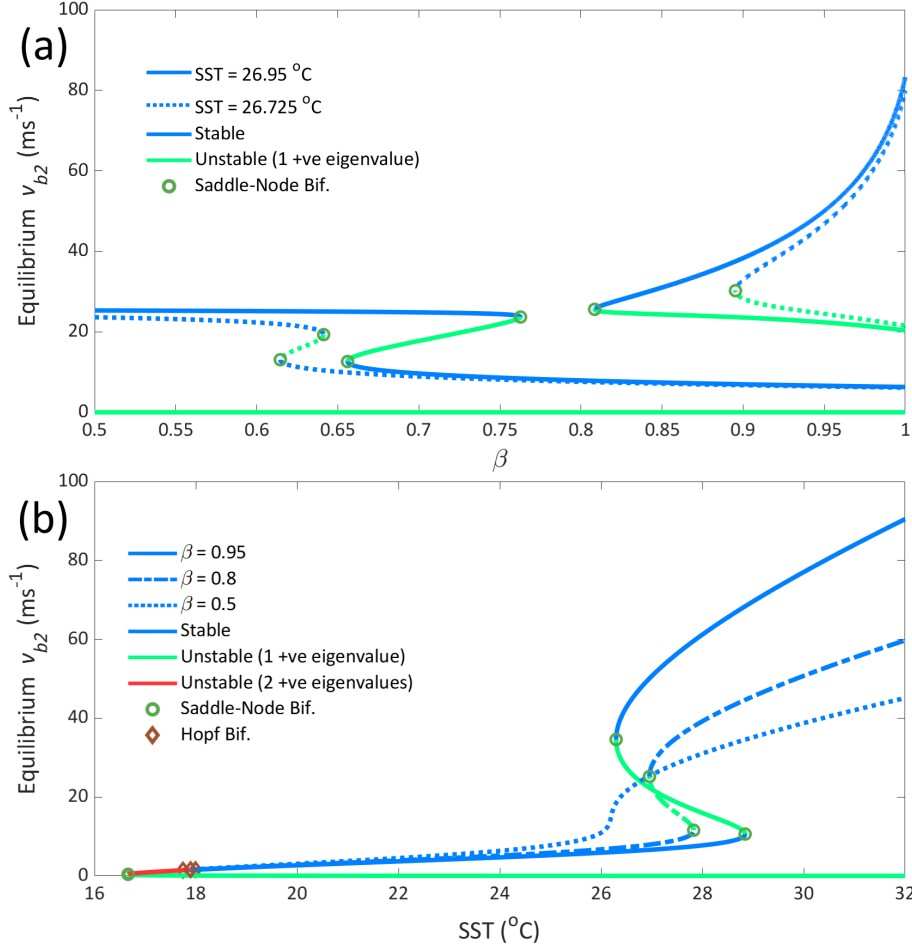

**Figure 2.** Bifurcation diagrams of model equilibria in a) $\beta$ and b) SST. Blue denotes stable equilibria, green unstable (one positive eigenvalue), red unstable (two positive eigenvalues), saddle-node bifurcations are marked as circles, and Hopf bifurcations are marked as diamonds. In a), for the dotted line SST $= 26.725^oC$ and for the solid line SST $= 26.95^oC$. In b), for the dotted line $\beta = 0.5$, for the dashed line $\beta = 0.8$, and for the solid line $\beta = 0.95$.

transitions could represent a few different phenomena. A transition from the low to high wind state can be interpreted as the intensification of a tropical depression into a TC and likewise, a transition from the high to low wind state as the dissipation of a TC into a tropical depression. A more nuanced phenomenon like an ERC will consist of a cycle of multiple, and possibly incomplete, transitions between these stable states.

## 3.2 Rate–Induced Behaviour in the Model

The effect of external forcing can be thought of as a shifting of the stability landscape while maintaining its qualitative features. If a tipping threshold, such as an unstable equilibrium, moves past the original position of a stable equilibrium of the unforced





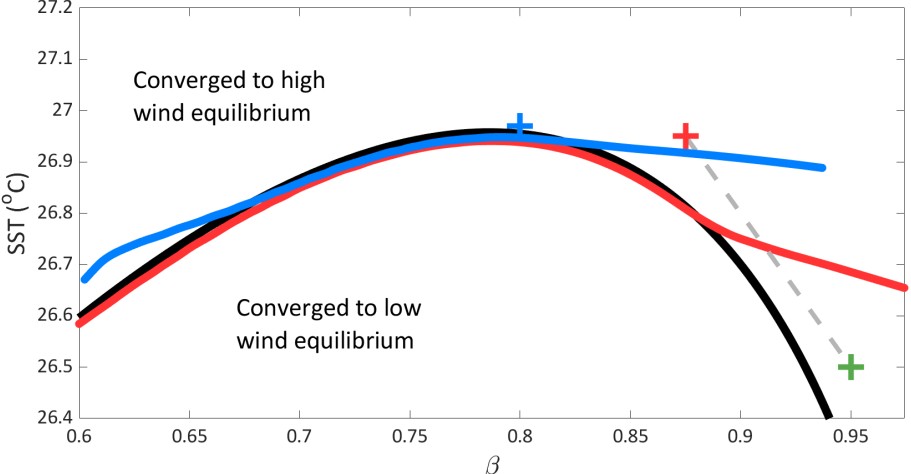

**Figure 3.** Examples of basin instability when considering instantaneous changes in $\beta$ and SST. The initial conditions are denoted by the blue ($\beta = 0.8$, SST $= 26.97\,^oC$) and red ($\beta = 0.875$, SST $= 26.95\,^oC$) crosses with their corresponding basin instability boundary in the same colour. The saddle–node continuation in $\beta$–SST, where the high wind equilibrium begins, is given by the black line. The initial conditions converge to the high wind equilibrium. Parameter conditions above the basin instability boundary lines re–converged to a high wind equilibrium after the instantaneous change and conditions below the line converged to a lower wind equilibrium. The green cross ($\beta = 0.95$, SST $= 26.5$) shows the conditions reached via forcing (profile trajectory shown in grey) in Figure 4.

system, this stable equilibrium is said to be threshold unstable to varying of the forcing rate (Wieczorek et al., 2023). When this threshold separates two stable equilibrium, the system is said to be *basin unstable*. In general, it has been shown that basin

instability is a sufficient condition for rate–induced tipping to occur, i.e. there exists some external forcing that will produce rate–induced tipping if the system is basin unstable (Wieczorek et al., 2023). In many examples it has been found that basin instability is both necessary and sufficient for rate–induced tipping to occur (Ritchie et al., 2023).

Here, we test for basin instability of the high wind state in the $\beta$–SST phase space. We first choose a $\beta$–SST point and find the high wind equilibrium corresponding to this parameter choice. The system is then integrated forward in time over a range

of fixed $\beta$ and SST values while using the original equilibrium value as the initial condition. If for a given $\beta$–SST combination the system remains in the a high wind state, this parameter pair is within the basin of attraction for that initial condition. Alternatively, if the system converges to a different equilibrium, here the low wind state, then the parameter pair is not within the basin of attraction for that initial condition. Two examples of basin instability for the model are shown in Figure 3. Here we choose an initial condition for the high wind state (the blue and red crosses) and test for tipping to the low wind state. The

red and blue curves are the basin instability boundaries corresponding to the respective initial condition. We see that the basin boundaries follow the saddle–node boundary for low and intermediate values of $\beta$, but for larger $\beta$ the two diverge. This area of divergence is of interest as is shows where rate–induced tipping may occur (as opposed to traditional bifurcation–induced tipping across the saddle–node).



Using the information provided by the basin instability diagram shown in Figure 3, we can produce examples of rate–induced tipping in the model. To define the evolution of a given parameter $\sigma$ with time we use a hyperbolic secant profile defined as

$$f(t; \lambda, P) = \pm \operatorname{sech}(\lambda(t - P)) + 1, \tag{9}$$

where $+$ gives an increasing and $-$ a decreasing profile, $\lambda$ determines the rate of change, and $P$ the time of peak forcing. This function was selected for its smooth transition between a defined maximum and minimum. Thus, for a given forcing parameter and maximum and minimum values $(\sigma_{min}, \sigma_{max})$ between which the forcing will occur, we define the time evolution of $\sigma$ as

$$\sigma(t; \lambda, P) = \sigma_{min} + (\sigma_{max} - \sigma_{min}) f(t; \lambda, P). \tag{10}$$

In cases where we require a strictly increasing or decreasing profile with no return to the original value, we define the evolution as

$$\sigma(t; \lambda, P) = \begin{cases} \sigma_{min} + (\sigma_{max} - \sigma_{min}) f(t; \lambda, P), & \text{if } t < P \\ \sigma_{min} \text{ or } \sigma_{max}, & \text{if } t > P. \end{cases} \tag{11}$$

We consider a parameter path outlined by the dotted line in Figure 3 with initial and final values given by the red and green crosses respectively. Figure 4 shows a time integration of the system and the forcing profiles applied. The forcing profiles in the two examples differ only in their rates, which where chosen to be on either side of the critical rate. It is also important to note that the forcing profiles never reach the critical values which would push the system across a bifurcation point (as shown in Figure 3). For the solid line profile, with rate $\lambda = 0.1$, we see the forcing is slow enough for the system to follow the high wind state, leading to an intensification of the TC. For the dotted line profile, with rate $\lambda = 0.3$, the forcing exceeds the critical rate and the system crosses the unstable mid wind state threshold, causing it to tip to the low wind state, leading to a dissipation of the TC.

In Figure 5 we provide some examples of overshoot recovery in the model. In Figure 5a, $\beta$ is forced with a return profile which crosses the saddle–node bifurcation. We see a very small change in the forcing rate can determine whether the model recovers to the high wind state or tips to the low wind state. Interestingly, the system appears to move to the unstable mid wind state for a considerable period of time before either recovering or tipping. The similarity between this behaviour and that observed in ERCs is discussed in Section 5. In Figure 5b, similar behaviour can be seen for forcing of SST, where a small change to the rate at which a temporary reduction in SST occurs can determine whether the system recovers or tips. Here however, instead of spending an intermediate period at the unstable mid wind state, as with the $\beta$ forcing, the system moves completely to the low wind state before beginning to recover, the success of which is dependent on the forcing rate.

## 4   Case Study: Hurricane Irma

Here we test the ability of the low–order model to produce realistic tangential wind profiles such as the one observed for Hurricane Irma (2017). Irma underwent two separate periods of rapid intensification (RI), the first shortly after its formation



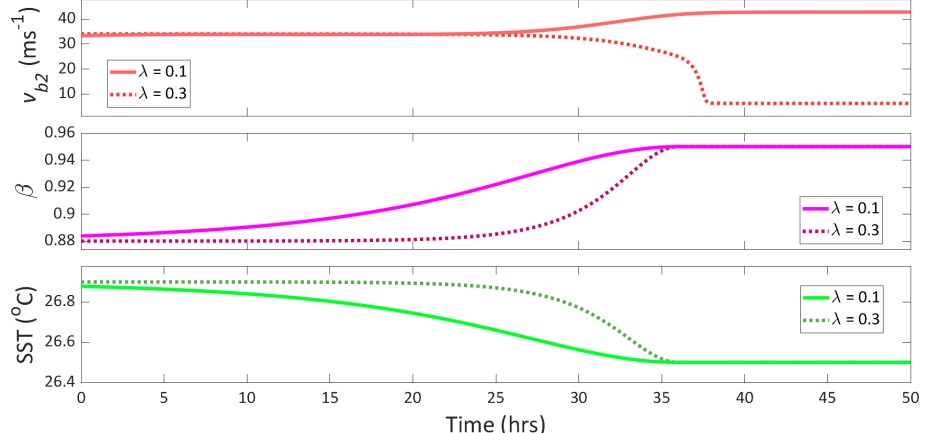

**Figure 4.** Example of the rate threshold between tracking and tipping when no bifurcation is crossed, with tangential wind speed and corresponding $\beta$ and SST forcing profiles. For both the solid line and dotted line profiles the maximum and minimum values of $\beta$ and SST are the same (decreasing ramp profiles with $0.875 \leq \beta \leq 0.95$ and $26.5 \leq SST \leq 26.95^{o}C$). The solid forcing profiles have a rate of $\lambda = 0.1$ and we see a tracking of the high wind equilibrium, whereas the dotted forcing profiles have a rate of $\lambda = 0.3$ and we see tipping to the low wind equilibrium.

and a second as it moved into the warmer waters around the edge of the Caribbean Sea (Fischer et al., 2020). In the context of this model, recreating the first RI period is of interest in relation to factors of TC intensification. The second is of interest in

relation to the dynamics of ERCs.

The first period of RI lasted for approximately two days from its time of formation. This RI period saw the maximum tangential wind increase by $33\,ms^{-1}$ from approximately $18\,ms^{-1}$ to $51\,ms^{-1}$ (Fischer et al., 2020). During this time Irma moved into an area of lower SST, meaning that a decreasing SST profile accompanies this RI period. In Figure 6a are the results of the recreation of this intensification period. The SST profile was estimated using the best track estimate latitude

and longitude coordinates for Irma along with daily gridded SST data from NOAA (Cangialosi et al., 2021). To initiate an intensification over this decreasing SST profile, we noted that the range over which SST is varying (27-28.5 $^{o}C$) coincides with the range within which a saddle–node bifurcation occurs in the low wind equilibrium (see Figure 2 b)). We thus applied a decreasing return (hyperbolic secant) $\beta$ forcing profile to tip the system over the low–wind–state bifurcation. Once the system crosses the bifurcation it transitions to the high wind equilibrium producing an intensification of the TC. The $\beta$ profile used in

Figure 6a has $\lambda = 0.15$. We found that an increase in $\lambda$ (e.g. $\lambda = 0.2$) enabled the system to regain the low wind equilibrium without tipping, i.e. no intensification occurred. As the system still crosses the bifurcation point, this is an example of the overshoot recovery discussed in Sect.3.2. Thus, in this context, we can describe the first RI as a 'missed' return tipping.

The second RI period began five days after its formation as it began to move over a region of increased SST. The RI period lasted two days and increased the maximum tangential wind speed up to $80\,ms^{-1}$. The second RI period was characterised

by two consecutive ERCs. The ERCs observed in Irma occurred over a much shorter time span than is typical of ERCs, each





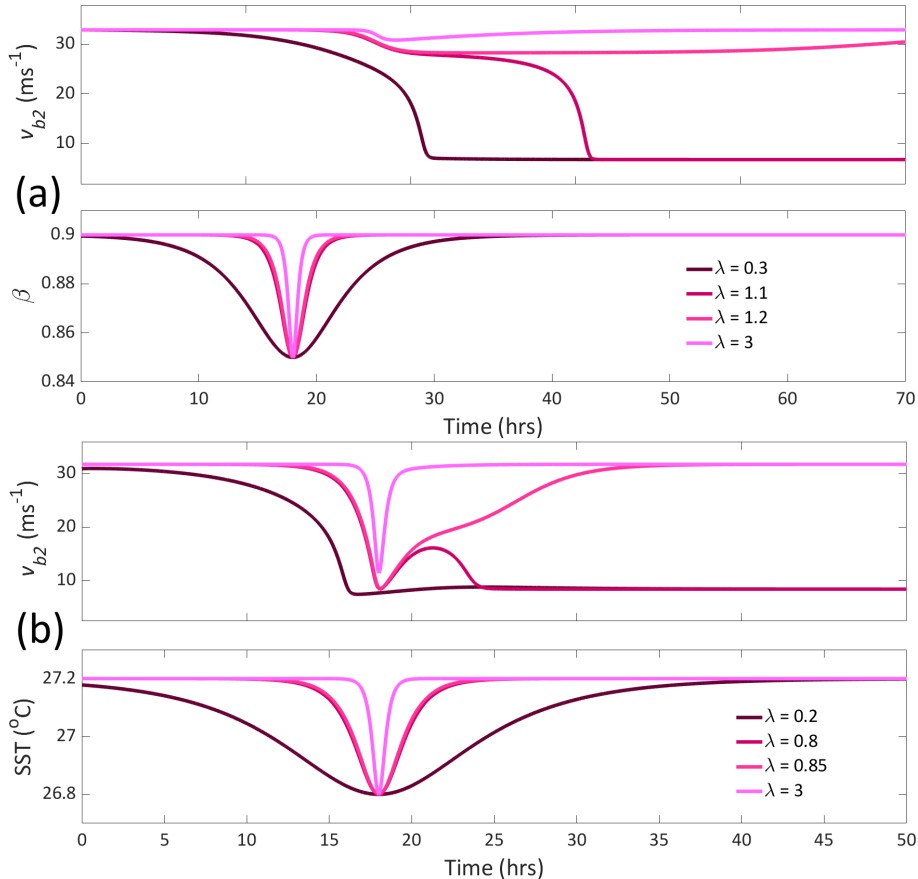

**Figure 5.** Example of the rate threshold between tipping and recovery when a bifurcation is crossed, with tangential wind speed and corresponding $\beta$ or SST forcing profiles. In a), $\beta$ is forced between the same maximum and minimum (decreasing return profile with $0.85 \leq \beta \leq 0.9$) with fixed SST $= 26.725^oC$ for a range of forcing rates. In b), SST is forced between the same maximum and minimum (decreasing return profile with $26.8 \leq SST \leq 27.2^oC$) with fixed $\beta = 0.8$ for a range of forcing rates.

taking around $10\,hr$ verses the average of $36\,hr$ (Sitkowski et al., 2011; Fischer et al., 2020). Although the wind profiles of the secondary eyewalls and the ERC event were not directly measured, Fischer et al. (2020) estimated a radial wind profile of the event using NOAA fly-through data. To recreate a similar scenario we forced the model with the estimated SST profile for the second RI period and a double–decreasing return $\beta$ profile. This resulted in an overall increase in tangential wind with two

isolated reductions corresponding to the change in $\beta$. When interpreting changes in the eyewall radius ($r_{b2}$) to be tracking the radius of maximum wind (RMW), the model produces a change in the RMW during the ERC events on a similar scale to that estimated by Fischer et al. (2020) (magnitude $\times 10^4\,m$).

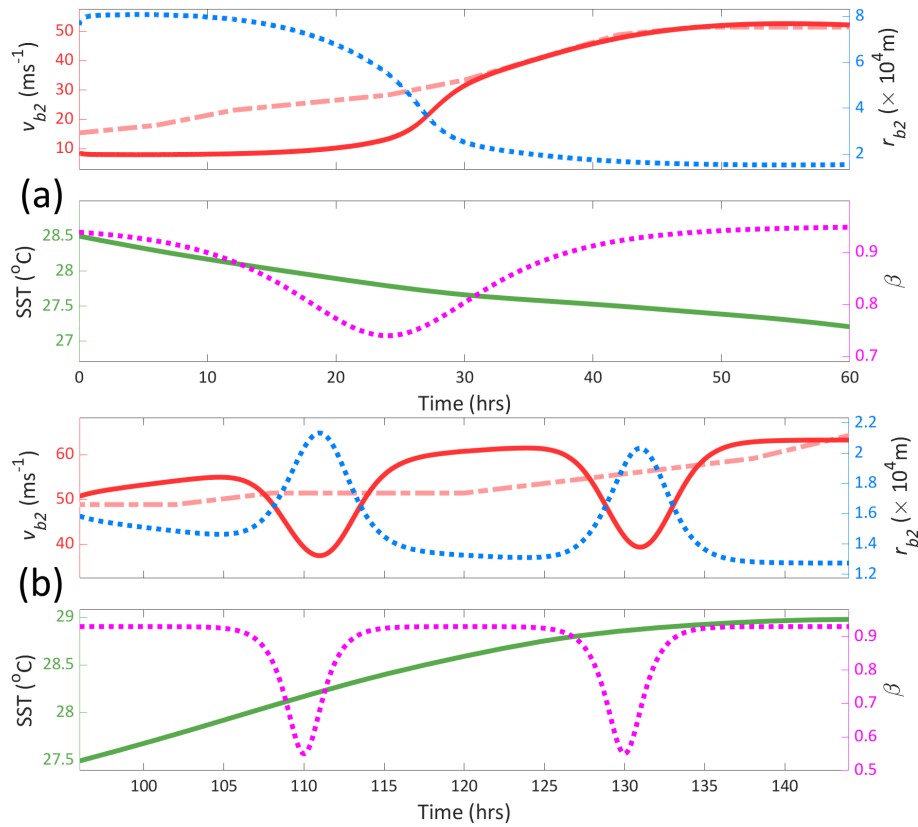

**Figure 6.** Comparison of tangential wind evolution between the model (red solid) and that observed for Hurricane Irma (light red dashed). The model is forced with an SST profile estimated using Irma best track data and daily gridded SST from NOAA (green solid) and a conceptual $\beta$ profile (purple dotted). The outer eyewall boundary layer radius ($r_{b2}$) is also shown (blue dotted). In a), the first RI period is modelled with a decreasing return $\beta$ profile ($0.74 \leq \beta \leq 0.95$, $\lambda = 0.15$) so as to initiate tipping to the high wind state. In b), the second RI period is modelled with a double decreasing $\beta$ profile ($0.55 \leq \beta \leq 0.93$, $\lambda = 1$) to recreated a double ERC event.

## 5 Discussion

It is informative to highlight the justification for the $\beta$ forcing profiles used in Sect. 3. We discussed the theories of Kepert
(2013) and Ge et al. (2015) regarding the role of the vorticity gradient in ERCs and intensification, outlined the inclusion of vorticity in the model, and described the connection between $\beta$ and the vorticity gradient via the tangential wind profile and radial inflow. Thus, interpreting a "dip" in $\beta$ as representing a change in the radial vorticity gradient follows directly from equation (8). We see that a decrease in $\beta$ increases the radial vorticity gradient at the outer eyewall boundary. This change is in line with both Kepert (2013) and Ge et al. (2015), where an increase in the radial vorticity gradient is responsible for an ERC
or intensification. As box models, such as this one, are not defined over a spatial domain, it is necessary to interpret changes in the radial vorticity gradient throughout the boundary layer as changes at the outer eyewall boundary. We acknowledge that



this does not allow for the definition of multiple radial vorticity gradient maxima as used by Kepert (2013) or the definition of differing inner–core vorticity profiles as used by Ge et al. (2015). Instead, this study has shown that at the level of a low–order conceptual model, a temporary increase in the radial vorticity gradient can initiate intensification (Fig. 6a), dissipation (Figs. 4, 5) and ERC–like behaviour (Fig. 6b).

The $\beta$ and SST forcing profiles used in this study can be likened to various physical situations. Return profiles, such as in Figure 5, model a temporary reduction and return of the parameter value. In the case of $\beta$, such a profile would be caused by a temporary restriction to the radial inflow of the TC. One interesting scenario where such variation in the radial inflow has been observed is in the diurnal variation of TC boundary layer flow (Zhang et al., 2020). These daily changes in the boundary layer produce return profiles in the radial inflow of similar shape and over similar time spans to those used in Figures 5. In the case of SST, return profiles represent the movement of the TC over an isolated area of cooler, or warmer (for an increasing return profile), SST. For example, the forcing profiles like the one used in Figure 5 could model a physical situation such as the movement of a TC over a region of upwelling of cooler water from the deep ocean as has been observed within TC regions (Park and Kim, 2010). The interaction of TCs with SST temperature profiles such as these has been observed to produce interesting behaviour, such as in the case of Tropical Cyclone Nari (2001) which moved back and forth across the Kuroshio current multiple times, causing its intensity to fluctuate. The rapid change in SST experienced by Nari as it oscillated across the warmer water of the Kuroshio and the cooler surrounding sea could have produced SST profiles similar to those used here. In comparison to return profiles, ramp profiles, such as in Figure 4, model gradual and continuing increases or decreases in the parameter values. These ramp profiles are useful for recreating the conditions often present during TC intensification as seen in Figure 6 where the estimated SST for Hurricane Irma produced a similarly shaped profile.

The discovery of rate–induced tipping in the low–order TC model suggests that external forcing rates play a role in TC dynamics. Very little research has been conducted into rate–induced phenomena in TCs. In a recent analysis of a low-order model representing TC formation, Slyman et al. (2023) identified rate–induced tipping by forcing two different parameters; the potential velocity and wind shear. These findings point to the possibility of rate–induced tipping pervading multiple aspects of TC dynamics.

We also found that the rate of forcing determines the systems ability to overshoot a bifurcation point but recover its original equilibrium. No previous research has been done into the temporary exceeding of critical parameter levels in TC models. In the case of this TC model, these brief parameter anomalies can have nice interpretations in terms of the movement of a TC through changing environmental conditions.

Observations of rate–dependent phenomena as described here have direct implications for TC prediction. Quantities such as a TC's tracking speed and the SST distribution in its path can be measured, thus allowing for approximations of rates of external forcing. Due to the conceptual nature of the model, we have focused on the qualitative behaviour that can result from different forcing rates. In order to make quantitative predictions about critical rates of forcing, further research of rate–induced tipping in higher complexity spatially–extended TC models coupled with more realistic forcing profiles will be needed.



The results of this study broaden our understanding of the role of the vorticity gradient as a driver of TC behaviour. They also expand upon the general dynamical properties of TCs. From these results it is clear there are potential advances to be made in TC modelling and prediction by further research in this area .

*Code availability.* All numerical computations for this study were performed in MATLAB R2022a. To compute solution trajectories of (3) we used the $4^{th}$–order Runge–Kutta finite–difference method. In order to perform a bifurcation analysis of the model we used continuation methods from the Continuation Core and Toolboxes (COCO) (Dankowicz and Schilder, 2013). The MATLAB code required to reproduce these results is available at https://doi.org/10.5281/zenodo.10846204.

**Appendix A: Supplementary Model Outline**

**A1  Auxiliary Equations**

A detailed derivation of these governing equations is provided by S&F (2012). Only an overview of the important components is presented here.

**A1.1  Mass-stream Function**

The Boussinesq approximation ensures non-divergence of the radial and vertical flow within the boundary layer and hence a mass-stream function for the boundary layer may be introduced as

$$\Psi_b = 2\pi r_b \rho_b C_D \frac{|v_b|v_b}{\zeta_b}, \tag{A1}$$

where the subscript $b$ denotes evaluation at $z = H_b$, $C_D$ is the transfer coefficient for momentum, and $\zeta$ is the absolute vorticity. For the purposes of this model the mass-stream function is only considered at the outer edge of the eyewall boundary layer, i.e. $\Psi_{b2}$.

**A1.2  Physical Radius and Tangential Velocity**

The model applies a version of the thermal wind balance equation derived by Emanuel (1986). Assuming gradient wind balance, saturated pseudoadiabatic ascent, and conservation of angular momentum, Emanuel takes the radial thermal wind balance and assumes the specific volume ($1/\rho$) to be expressible as a function of pressure and saturated entropy. Coupled with the assumption that the saturated entropy does not vary along surfaces of equal angular momentum, this allows the thermal wind balance to be expressed as a relationship between (specific) saturated entropy, $s^*$, and the angular momentum per unit mass, $m$:

$$\frac{T_b - T}{m}\frac{ds^*}{dm} = 2\frac{T_b - T}{f^2 R^3}\frac{ds^*}{dR} = \frac{1}{r^2} - \frac{1}{r_b^2}, \tag{A2}$$





where $T$ denotes temperature, $r \equiv r(z; R)$ is the physical radius of a given potential radius, and $r_b \equiv r_b(R)$ is the physical radius in the boundary layer corresponding to the potential radius. This balance relates the angular momentum surfaces to the potential radius and the change in saturated entropy with potential radius. Thus, equations for the evolution of the physical radii of the inner and outer edges of the eyewall boundary layer are found by taking $R = R_1$ and $R = R_2$ and approximating

the change in saturated entropy (via finite difference). For closure the mass, $M$, enclosed by the angular momentum surface at $R = R_2$ is assumed to be conserved (boundary layer inflow matches troposphere outflow) and as the eye is modeled by solid body rotation its mass, $M_e$, enclosed by the angular momentum surface at $R = R_1$ will also be conserved. Using the Boussineq approximation of near constant density these masses can be found (derived by Frisius (2005)) as

$$M = \pi \rho \int_{H_b}^{H+H_b} r_2^2 \, dz = \frac{\pi \rho}{G_2} ln \left( 1 + G_2 r_{b2}^2 H \right) \tag{A3a}$$

and $\quad M_e = \pi \rho \int_{H_b}^{H+H_b} r_1^2 \, dz = \frac{\pi \rho}{G_1} ln \left( 1 + G_1 r_{b1}^2 H \right), \tag{A3b}$

where $r_1$ and $r_2$ are the physical radii of the angular momentum surfaces at $R = R_1$ and $R = R_2$ respectively, and $r_{b1}$ and $r_{b2}$ are the physical radii of the inner ($R = R_1$) and outer ($R = R_2$) edges of the eyewall boundary layer. The functions $G_1$ and $G_2$ are given as

$$G_2(s_i^*) = \frac{2\Gamma}{f^2 R_2^3} \frac{s_a^* - s_i^*}{\Delta R}, \quad G_1(s_i^*) = \frac{2\Gamma}{f^2 R_1^3} \frac{s_a^* - s_i^*}{\Delta R} \left( \frac{R_1}{R_2} \right)^{\kappa-1}, \tag{A4}$$

where $\Gamma$ is the temperature lapse rate, $f$ is the Coriolis parameter, and $\kappa$ is called the eyewall entropy profile parameter and controls the radial decrease in saturated entropy away from the radius of maximum wind at $R_2$. The temperature lapse rate controls the vertical temperature profile which in this model is taken to be linear and defined as

$$\Gamma = \frac{T_s - T_t}{H}, \tag{A5}$$

where $T_t$ is the tropopause temperature, $T_s$ is the sea surface temperature, and $H$ is the tropopause height. The mass equations
(A3) can then be rearranged to find $r_{b1}$ and $r_{b2}$ as

$$r_{b2}(s_i^*) = \sqrt{\frac{1}{G_2 H} \left[ \exp \left( \frac{G_2 M}{\pi \rho} \right) - 1 \right]}, \quad r_{b1}(s_i^*) = \sqrt{\frac{1}{G_1 H} \left[ \exp \left( \frac{G_1 M_e}{\pi \rho} \right) - 1 \right]}. \tag{A6}$$

and using (2) the corresponding tangential wind speeds at these points can be found as

$$v_{b2}(s_i^*) = \frac{f}{2} \left( \frac{R_2^2 - r_{b2}^2}{r_{b2}} \right), \quad v_{b1}(s_i^*) = \frac{f}{2} \left( \frac{R_1^2 - r_{b1}^2}{r_{b1}} \right). \tag{A7}$$

**A1.3 Mass**

As the eye and eyewall mass ($M_e$, $M_i$, $M = M_e + M_i$) are assumed to be conserved they can be calculated for the resting state when the eyewall boundaries are assumed to be vertically oriented, i.e. $r_1 = R_1$ and $r_2 = R_2$. They are then

$$M_e = \pi \rho H R_1^2, \quad M = \pi \rho H R_2^2, \quad M_i = M - M_e = \pi \rho H \left( R_2^2 - R_1^2 \right). \tag{A8}$$



The masses of the boundary layer beneath the eyewall ($M_{bi}$) and the ambient region ($M_{ba}$) are not assumed to be conserved and are given by

$$M_{bi}(s_i^*) = \pi \rho_b H_b \left( r_{b2}^2 - r_{b1}^2 \right), \quad M_{ba}(s_i^*) = \pi \rho_b H_b \left( r_{ba}^2 - r_{b2}^2 \right), \tag{A9}$$

where $r_{ba}$ is the outer radius of the ambient region.

### A1.4 Entropy

The entropy of the sea surface underneath the eyewall region is taken to be

$$s_{oi}(s_i^*) = L_v \left( \frac{q_v^* - q_{v,ref}}{T_s} \right) + \frac{v_{b2}^2}{2 T_s \beta} \left[ 1 - \left( \frac{r_{b2}}{r_a} \right)^{2\beta} \right] - \frac{f v_{b2} r_{b2}}{T_s (1-\beta)} \left[ 1 - \left( \frac{r_a}{r_{b2}} \right)^{1-\beta} \right], \tag{A10}$$

where $L_v$ is the latent heat of vaporisation, $q_v^*$ is the specific humidity at saturation, and $q_{v,ref}$ is the reference specific humidity. The entropy of the sea surface far from the TC ($R \to \infty$) is taken as

$$s_{oa0} = L_v \left( \frac{q_v^* - q_{v,ref}}{T_s} \right), \tag{A11}$$

and the sea surface entropy under the ambient region is taken as the average of these two as

$$s_{oa}(s_i^*) = \frac{s_{oi} + s_{oa0}}{2}. \tag{A12}$$

The entropy of the ambient region itself is taken to be

$$s_a = L_v \left( \frac{q_{v,a}}{T_a} - \frac{q_{v,ref}}{T_{ref}} \right) - R_d \ln \left( \frac{p_a}{p_{ref}} \right) + c_p \ln \left( \frac{T_a}{T_{ref}} \right), \tag{A13}$$

where $q_{v,a}$ is the specific humidity of the ambient region, $T_{ref}$ is a reference temperature, $R_d$ is the specific gas constant of dry air, $p_a$ is pressure of the ambient region, $p_{ref}$ is a reference pressure, $c_p$ is the specific heat of dry air at constant pressure, and $T_a$ is the temperature of the ambient region defined as

$$T_a = T_s \left( \frac{p_a}{p_{ref}} \right)^{\frac{R_d \Gamma}{g}}. \tag{A14}$$

For the saturated entropy of the ambient region, $s_a^*$, the specific humidity is taken at saturation ($q_{v,a}^*$ instead of $q_{v,a}$).

### A2 Constant Parameter Values

The model parameters used by S&F are given in Table A1.

### A3 Specific Humidity

S&F do not provide values for the specific humidities $q_v^*$, $q_{v,a}$, $q_{v,a}^*$ and $q_{v,ref}$. To calculate the specific humidity at saturation from the air temperature we fit a two–term exponential model to experimental data (ToolBox, 2009) (exp2 function in MATLAB) which resulted in

$$q(T) = 1.445 \times 10^{-6} e^{0.221 T} + 4.967 e^{5.718 \times 10^{-2} T} \tag{A15}$$



| Notation | Value | Description |
|----------|-------|-------------|
| $r_{ba}$ | $420\,km$ | Outer radius of ambient region |
| $\tau_E$ | $48\,h$ | Timescale of diabatic cooling |
| $\tau_C$ | $4\,h$ | Timescale of convective exchange |
| $C_H$ | $0.003$ | Transfer coefficient for enthalpy |
| $C_D$ | $0.003$ | Transfer coefficient for momentum |
| $H$ | $13.5\,km$ | Difference between tropopause and boundary layer heights |
| $H_b$ | $1.5\,km$ | Boundary layer height |
| $f$ | $5 \times 10^{-5}\,s^{-1}$ | Coriolis parameter |
| $\kappa$ | $3$ | Eyewall entropy profile parameter |
| $R_1$ | $90\,km$ | Inner potential radius of eyewall |
| $R_2$ | $180\,km$ | Outer potential radius of eyewall |
| $\Delta R$ | $30\,km$ | Distance from eyewall to outer region |
| $\rho$ | $0.45\,kg\,m^{-3}$ | Mean density |
| $\rho_b$ | $1.1\,kg\,m^{-3}$ | Mean boundary layer density |
| $T_t$ | $203.15\,K$ | Tropopause temperature |
| $T_s$ | $301.15\,K$ | Sea surface temperature |
| $p_a$ | $500\,hPa$ | Ambient region pressure level |
| $p_{ref}$ | $1000\,hPa$ | Reference surface pressure |
| $h_{b,ref}$ | $80\%$ | Boundary layer relative humidity |
| $h_a$ | $45\%$ | Ambient region relative humidity |
| $\delta$ | $0.25$ | Entrainment parameter |
| $\beta$ | $0.875$ | Tangential wind profile parameter |

**Table A1.** Model parameters given by S&F (Schönemann and Frisius, 2012)

where $T$ is the temperature in degrees. Then $q_v^*$ is calculated at $T_s$ and $q_{v,a}^*$ at $T_a$. The non–saturated humidities are then

calculated as $q_{v,ref} = h_{b,ref} q_v^*$ and $q_{v,a} = h_a q_{v,a}^*$. When comparing the function output with observational data of specific humidity gathered during TC season (Jordan, 1958), values for $q_{v,ref}$ were close to those observed but the values of $q_{v,a}$ were smaller than observed thus we scaled $q_{v,a}^*$ by a factor of 1.7 to match with observations. We assume this discrepancy is a result of (A15) considering only the temperature difference and not the pressure difference between the two regions. We also take the reference temperature $T_{ref}$ to be equal to the sea surface temperature $T_s$.

*Author contributions.* Both authors contributed to the design of the study and the writing of the manuscript. S.W. undertook the analysis as part of an Honours thesis with C.Q. as supervisor.



*Competing interests.* The authors declare there are no competing interests.

*Acknowledgements.* This work constitutes part of the work conducted by the first author during his Honours year at the University of Tasmania. Thanks also to Paul Ritchie and Hassan Alkhayuon for useful comments and advice.



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
