# Peer review of "The Role of the Radial Vorticity Gradient in Intensification of Tropical Cyclones"

_EGUsphere, 2024_

## Author Comment (AC2)

The Role of Radial Vorticity Gradient in the Intensification of Tropical Cyclones
Courtney Quinn
Samuel Watson

**Response to reviewer comments**

Dear Reviewers,

We would like to thank you both for your thorough reviews of our manuscript. We feel that your comments prompted insightful discussion that has improved our manuscript. Below we include our responses to your comments in blue. We have also made adjustments to the manuscript where necessary and have included our revised manuscript with tracked changes.

Many thanks,

Courtney Quinn and Samuel Watson
* * *
**REVIEW 1 - Lin Li**

In the paper "The Role of Radial Vorticity Gradient in the Intensification of Tropical Cyclones," the authors use a three-variable model to treat tropical cyclones (TCs) as a dynamical system, with which they explore the role of SST and vorticity gradient in the stable states of the model and examine how changes in these variables can cause rate-induced tipping. This paper provides a fresh perspective on the study of TC intensification and is worth publication after fixing the following issues.

We thank you for the positive evaluation.

I have the following questions and comments regarding the simple model:

1. The construction of this model is quite interesting. However, the enthalpy transport from the ocean to the boundary layer between $r_{b1}$ and $r_{b2}$ is not considered in the model. The enthalpy transfer in this region could contribute significantly to the total enthalpy transfer (ref 1). I hope the authors can explain why this part was excluded or how neglecting it might affect the results.

In equation 3b, the second term (that which is multiplied by $C_H$, the surface transfer coefficient for enthalpy) represents the surface transfer of latent heat into the boundary layer between $r_{b1}$ and $r_{b2}$. Here, $s_{oi}(s_i^*)$ is the mean entropy at the ocean surface beneath the eyewall. This is a function of $s_i^*$ implicitly as it depends on the radial extent of the eyewall $(r_{b2}(s_i^*))$. In order to model the overall amount of enthalpy transfer in the eyewall boundary layer from the ocean, the magnitude of the surface

wind speed is taken as an average of the tangential wind speed between the surface boundaries of the eyewall ($v_{b1}$ and $v_{b2}$).

In order to clarify this contribution in the text, we have amended the sentence in line 122 to read "The second term gives the enthalpy transport from the ocean to the boundary layer between $r_{b1}$ and $r_{b2}$, which is estimated by the transfer of latent heat from the sea surface proportional to the average wind speed in the eyewall boundary layer".

2. The four equilibrium states (unstable no wind, stable low wind, unstable mid wind, and stable high wind) are a distinguishing result of this model. My question is whether the stable low-wind state is detectable in TC simulations. Here I direct the authors to consider (ref 2), in which Figure 2 shows TC intensity jumping between two states, implying the existence of two stable states rather than the traditionally thought one stable state. Successfully linking the model with existing TC simulations could strengthen this paper.

We thank you for directing our attention to this reference. The results do indeed suggest the existence of a stable low-wind state in CM1 (Bryan and Fritsch, 2002) which is commonly used to simulate TC behaviour. We have added a paragraph at the start of the discussion to emphasize the existence of such a state in higher complexity models and that our results can inform future studies of tipping behaviour in more realistic simulations.

Regarding rate-induced tipping:

1. The authors claim in the introduction that they use the model to explore eyewall replacement cycles (ERC). This should be approached with caution, as the model only includes wind speeds at two locations and is therefore unable to reveal multiple wind maxima in ERC. The wording should reflect this limitation to avoid overstatement.

You are correct in that the model cannot explicitly capture ERCs as it cannot identify multiple wind maxima. The ERCs are thus deduced from a combination of intensification of the TC with an increase in the radius of the outer eyewall boundary - this is now clarified in Section 2 where we discuss the model maximum wind. We have modified a sentence in the introduction to state that we focus on intensification and dissipation, and then relate the findings to ERCs through the associated dynamical parameters and the radial extent of the eyewall. We have also softened our statement in the discussions of the findings, noting that the model

suggests the existence of ERC-like behaviour rather than demonstrating it.

2. Although the title is "The Role of Radial Vorticity Gradient...," the focus of the paper seems to be on rate-induced tipping by various parameters (including vorticity gradient, SST, and possibly others) rather than the role of vorticity gradient itself. I suggest rephrasing the title and abstract to better reflect the real focus of the paper.

We agree with you that the title and abstract put more focus on one parameter than the other which were also considered. We have thus changed the title to read "The Role of Time-varying External Factors in the Intensification of Tropical Cyclones". We have also amended the abstract to reflect this focus and adjusted a few phrases in the introduction.

3. To make this paper more attractive to general readers, I suggest the authors add a conceptual figure explaining rate-induced tipping in TC rapid intensification, similar to Figure 1 of ref 3 but using the states of their model. This will make the paper more understandable to readers who are not familiar with rate-induced tipping.

Thank you for this suggestion. We have obtained permission from the authors of [3] to recreate their diagram for our model. Our schematic is included as Figure 3 in the updated manuscript and is introduced alongside the concept of basin instability in the text (Section 3.2).

**REVIEW 2 - Satoki Tsujino**

Summary:

The authors focused on the role of the radial vorticity gradient in tropical cyclone (TC) dynamics with a low–order conceptual model. In this study, for two parameters of the vortex shape in the storm outer radii and sea surface temperature, transitions from a stable state to another stable state (linked to intensification or dissipation in TC) were examined. They found rate-dependent behavior in the simple model framework by changing the two parameters.

General comments:

I think that the manuscript is well-organized in each part and essential behaviour of the intensity changes in TC is well captured by the simple model framework with external forcing. As mentioned by the authors, there are few researches on dynamical systems such as the present manuscript in TC literatures. Thus, the authors' work can potentially contribute to update and improvement of the understanding of dynamics and intensity changes in TCs. I recommend it is enough for publishing after minor revision.

We thank you for the supportive review of our manuscript.

Minor comments:

Equation (8): The vorticity gradient is defined as the $r_{b2}$ derivative of the $\zeta_{b2}$. However, $r_{b2}$ is one point value (not continuous valiable). I consider the definition may be simply $\partial\zeta/\partial r$. Please clarify it.

In this case, although $r_{b2}$ is defined as the outer eyewall boundary, it is actually a function of entropy (see Eq. A6). Thus the partial derivative with respect to $r_{b2}$ can be defined. We have clarified this in the text and made reference to the function for $r_{b2}$.

L315-317: The phrase "(boundary layer inflow matches troposphere outflow)" may be confused in readers. Exactly, the vertical integral of the lateral mass flux in the boundary layer is identical to that in the troposphere outflow layer. However, the speed of the boundary layer inflow is not identical to the speed of the troposphere outflow. I recommend that the phrase may need to be deleted.

We have removed this phrase to avoid confusion.

Equation (A15): The symbol of $q*$ is better that $q$ because the saturation is indicated by the asterisk.

Thank you for catching this oversight. We indeed meant this equation to represent the $q^*$ quantities. This has been adjusted in the revised manuscript.

L364: Is the unit of T (temperature) "degrees Celsius"? Please clarify it.

Yes, the unit is in degrees Celsius. We have added this clarification to the text.
* * *
**SUMMARY OF MAJOR MANUSCRIPT CHANGES**
"pg" refers to page number in following latexdiff document

| Title and abstract | pg 1 | Edited to better reflect focus of paper |
|---|---|---|
| Section 1 | pg 1 | Clarified connection of our results to eyewall replacement cycles (ERCs) |
| Section 2 | pg 6 | Explanation of how an ERC can be deduced from the low-order model |
| | | Clarification of $r_{b2}$ as a variable rather than a single point |
| Section 3 | pg 8 | New Figure 3 - schematic diagram of basin instability |
| Section 5 | pg 12 | Discussion of multiple stable states, including a low wind state, in more complex models |

**References**

[revised manuscript text omitted]